# Creation of an ustekinumab external control arm for Crohn's disease using electronic health records data: A pilot study

Vivek A. Rudrapatna[1,2]*, Yao-Wen Cheng[1], Colin Feuille[1], Arman Mosenia[3], Jonathan Shih[4], Yongmei Shi[2], Olivia Roberson[1], Benjamin Rubin[1], Atul J. Butte[2,5], Uma Mahadevan[1], Nicholas Skomrock[6], Ngozi Erondu[6], Christel Chehoud[6], Saquib Rahim[6], David Apfel[6], Mark Curran[6], Najat S. Khan[6], Christopher O'Brien[6], Natalie Terry[6], Benjamin D. Martini[6]

1 Division of Gastroenterology, Department of Medicine, University of California, San Francisco, San Francisco, CA, United States of America, 2 Bakar Computational Health Sciences Institute, University of California, San Francisco, San Francisco, CA, United States of America, 3 School of Medicine, University of California, San Francisco, San Francisco, CA, United States of America, 4 Department of Neurology, University of California, San Francisco, San Francisco, CA, United States of America, 5 Department of Pediatrics, University of California, San Francisco, San Francisco, CA, United States of America, 6 Janssen Research and Development, Spring House, PA, United States of America

* vivek.rudrapatna@ucsf.edu

**Data Availability Statement:** A deidentified version of the study data are being included as a part of this submission (supplemental data).

## Abstract

### Background

Randomized trials are the gold-standard for clinical evidence generation, but they can sometimes be limited by infeasibility and unclear generalizability to real-world practice. External control arm (ECA) studies may help address this evidence gaps by constructing retrospective cohorts that closely emulate prospective ones. Experience in constructing these outside the context of rare diseases or cancer is limited. We piloted an approach for developing an ECA in Crohn's disease using electronic health records (EHR) data.

### Methods

We queried EHR databases and manually screened records at the University of California, San Francisco to identify patients meeting the eligibility criteria of TRIDENT, a recently completed interventional trial involving an ustekinumab reference arm. We defined timepoints to balance missing data and bias. We compared imputation models by their impacts on cohort membership and outcomes. We assessed the accuracy of algorithmic data curation against manual review. Lastly, we assessed disease activity following treatment with ustekinumab.

### Results

Screening identified 183 patients. 30% of the cohort had missing baseline data. Nonetheless, cohort membership and outcomes were robust to the method of imputation. Algorithms for ascertaining non-symptom-based elements of disease activity using structured data were

**Funding:** Research reported in this publication was supported by Janssen Research and Development LLC, the UCSF Bakar Computational Health Sciences Institute, and the National Center for Advancing Translational Sciences, National Institutes of Health, through UCSF-CTSI Grant Number UL1 TR001872. Its contents are solely the responsibility of the authors and do not necessarily represent the official views of the NIH. The funder Janssen Pharmaceuticals provided support in the form of salaries for authors NS, NE, CC, SR, DA, MC, NSK, CO, NT, and BDM, but did not have any additional role in the study design, data collection and analysis, decision to publish, or preparation of the manuscript. The specific roles of these authors are articulated in the 'author contributions' section.

**Competing interests:** NS, NE, CC, SR, DA, MC, NSK, CO, NT, and BDM are employees of Janssen Pharmaceuticals, a for-profit entity that owns all rights to the drug ustekinumab. UM is a consultant for Janssen Pharmaceuticals. AJB is a co-founder and consultant to Personalis and NuMedii; consultant to Mango Tree Corporation, and in the recent past, Samsung, 10x Genomics, Helix, Pathway Genomics, and Verinata (Illumina); has served on paid advisory panels or boards for Geisinger Health, Regenstrief Institute, Gerson Lehman Group, AlphaSights, Covance, Novartis, Genentech, and Merck, and Roche; is a shareholder in Personalis and NuMedii; is a minor shareholder in Apple, Meta (Facebook), Alphabet (Google), Microsoft, Amazon, Snap, 10x Genomics, Illumina, Regeneron, Sanofi, Pfizer, Royalty Pharma, Moderna, Sutro, Doximity, BioNtech, Invitae, Pacific Biosciences, Editas Medicine, Nuna Health, Assay Depot, and Vet24seven, and several other non-health related companies and mutual funds; and has received honoraria and travel reimbursement for invited talks from Johnson and Johnson, Roche, Genentech, Pfizer, Merck, Lilly, Takeda, Varian, Mars, Siemens, Optum, Abbott, Celgene, AstraZeneca, AbbVie, Westat, and many academic institutions, medical or disease specific foundations and associations, and health systems. Atul Butte receives royalty payments through Stanford University, for several patents and other disclosures licensed to NuMedii and Personalis. Atul Butte's research has been funded by NIH, Peraton (as the prime on an NIH contract), Genentech, Johnson and Johnson, FDA, Robert Wood Johnson Foundation, Leon Lowenstein Foundation, Intervalien Foundation, Priscilla Chan and Mark Zuckerberg, the Barbara and Gerson Bakar Foundation, and in the recent past, the March of Dimes, Juvenile Diabetes Research Foundation, California Governor's Office of

accurate against manual review. The cohort consisted of 56 patients, exceeding planned enrollment in TRIDENT. 34% of the cohort was in steroid-free remission at week 24.

## Conclusion

We piloted an approach for creating an ECA in Crohn's disease from EHR data by using a combination of informatics and manual methods. However, our study reveals significant missing data when standard-of-care clinical data are repurposed. More work will be needed to improve the alignment of trial design with typical patterns of clinical practice, and thereby enable a future of more robust ECAs in chronic diseases like Crohn's disease.

## Introduction

The term *external control arm* (ECA) commonly refers to the use of observational cohorts to estimate treatment effects via indirect comparisons to other cohorts, particularly prospective ones participating in interventional trials. Recent years have seen a growing interest in constructing ECAs and analyzing their outcomes for a variety of purposes. A primary use case for these studies is in evaluating the applicability of findings made in controlled settings to that of the general populations and practices that typify routine clinical care. Another important use case has been in the context of regulatory approvals [1, 2], particularly in oncology and rare diseases where prospective trials are not always feasible [3]. A notable example of this occurred in 2019, when the FDA expanded the approved indications of a breast cancer drug, palbociclib, to also include men. This label expansion was approved on the basis of research that used electronic health records as well as medical claims databases [4].

Generally, ECAs have been performed in contexts where treatments and outcomes are relatively well-defined, and thus align well with prospective studies. However, the feasibility and robustness of ECAs for common, chronic, and complex diseases such as inflammatory bowel disease (IBD) remain unknown. There is an unmet need to better understand ECAs and evaluate their viability as a complement to prospective studies. Our own prior work has used electronic health records (EHR) data to estimate the effectiveness of tofacitinib for treating IBD [5]. While this study suggested that EHR data may be useful for this purpose, it was conducted on a cohort whose baseline characteristics substantially differed from the cohorts under study in the pre-approval trials of this drug.

The primary objective of this pilot study was to develop a method for creating an ECA for Crohn's disease. The target of our retrospective emulation was the ustekinumab comparator arm of TRIDENT, a recently completed phase 2b interventional trial of a new treatment for Crohn's disease [6]. Secondary objectives included 1) evaluating EHR-based algorithms as an alternative to manual ascertainment of some disease variables, 2) assessing the robustness of imputation methods for handling missing data, and 3) quantifying outcomes following treatment with ustekinumab.

## Materials and methods

### Ethics

Approved by the University of California, San Francisco (UCSF) Institutional Review Board (#20–31760). This board waived the requirement for informed consent, given the retrospective nature of this study.

Planning and Research, California Institute for Regenerative Medicine, L'Oreal, and Progenity. These commercial affiliations does not alter our adherence to PLOS ONE policies on sharing data and materials.

## Eligibility

This study was designed to retrospectively emulate a prospective cohort from TRIDENT, a recently completed phase 2b randomized controlled trial of a new potential treatment for Crohn's disease. The design of this trial included an active comparator arm consisting of patients randomized to receive ustekinumab, an FDA-approved treatment for this disease. We began by identifying eleven major eligibility criteria from the TRIDENT protocol, and adapting them to be retrospectively applied to UCSF patients who received ustekinumab as a part of the standard of care. These criteria select for adults with a Crohn's Disease Activity Index (CDAI) between 220 and 450 and who were stably exposed to other treatments for Crohn's disease. The CDAI is a composite score that ranges from 0 to over 600, and includes physical exam findings, laboratory measurements, and symptoms (i.e. patient reported outcomes). See S2 in S1 File for more details.

## Inclusion criteria

1. Age $\geq$ 18

2. A diagnosis of Crohn's disease as determined by the treating clinician

3. Receipt of ustekinumab at the FDA-approved route and dose

4. A baseline Crohn's Disease Activity Index (CDAI) within the range of 220 to 450 inclusive

## Exclusion criteria

1. Receipt of IV steroids within the 3 weeks prior to starting ustekinumab

2. A change to the dose of an oral corticosteroid (e.g. prednisone, budesonide) within the 3 weeks prior to starting ustekinumab

3. A change in the dose of methotrexate, 6-mercaptopurine, or azathioprine within the 12 weeks prior to starting ustekinumab

4. A change in the dose of antibiotics as used to treat Crohn's disease within the 3 weeks prior to starting ustekinumab

5. A change in the dose of 5-ASA compounds within the 3 weeks prior to starting ustekinumab

6. The occurrence of bowel surgery (segmental colectomy, ileocecectomy) within the 6 months prior to starting ustekinumab

7. A history of major bowel surgery (resection of >50% of the colon, multiple small bowel surgeries) at any point in time prior to starting ustekinumab

These correspond to the major criteria from TRIDENT, except for the exclusion of patients with recent exposure to tumor necrosis factor inhibitors (8 weeks) and integrin inhibitors (16 weeks). We did this for two reasons: 1) real-world patients who fail to benefit from one biologic are typically switched to another with minimal delay to avoid the risk of a flare, and 2) patients who sustained long biologic washout periods without being hospitalized or meeting other exclusionary criteria tend to have a lower disease severity and thus tended to be excluded due to a low baseline CDAI. This deviation from the emulation plan was not anticipated or planned a priori, but rather was made ex post after we recognized that the application of these criteria would have removed nearly all the otherwise eligible real-world patients from this cohort study.

## Cohort identification

We performed cohort identification in six sequential phases (Fig 1).

**Phase 1.** We used a database of structured EHR data at UCSF (2012–2020) to screen patient records meeting the following two criteria:1) at least one Crohn's disease diagnosis code (ICD-9-CM 555*; ICD-10-CM K50*) and 2) at least one medication order for ustekinumab. We have previously used this database to conduct studies of real-world treatment outcomes in IBD [5].

**Phase 2.** We equally divided the identified records among four trained chart abstractors to perform additional screening manually. The target of this first pass screening were 1) confirmation of Crohn's disease as documented by the treating clinician, 2) treatment with ustekinumab at the FDA-approved dose and route, and 3) all inclusion and exclusion criteria as listed above except that pertaining to the CDAI. Abstractors followed a detailed protocol to abstract these elements from the EHR (see S3 in S1 File), and evaluated all sections including labs performed outside UCSF and scanned documents from referring physicians. All chart abstraction was done under the supervision of the principal investigator, a gastroenterologist. Edge cases were adjudicated during weekly meetings.

**Phase 3.** At the end of phase 2 we had identified patients meeting all study criteria except for the one pertaining to the baseline CDAI. From this cohort, we abstracted the baseline patient-reported outcome elements of the CDAI (abdominal pain, diarrhea, wellbeing; PRO3) using a time window of up to 16 weeks prior to the date of the first dose of ustekinumab. We analyzed the data to empirically define the baseline period. This corresponded to the narrowest window of time prior to week 0 that was not associated with a substantial increase in missing data.

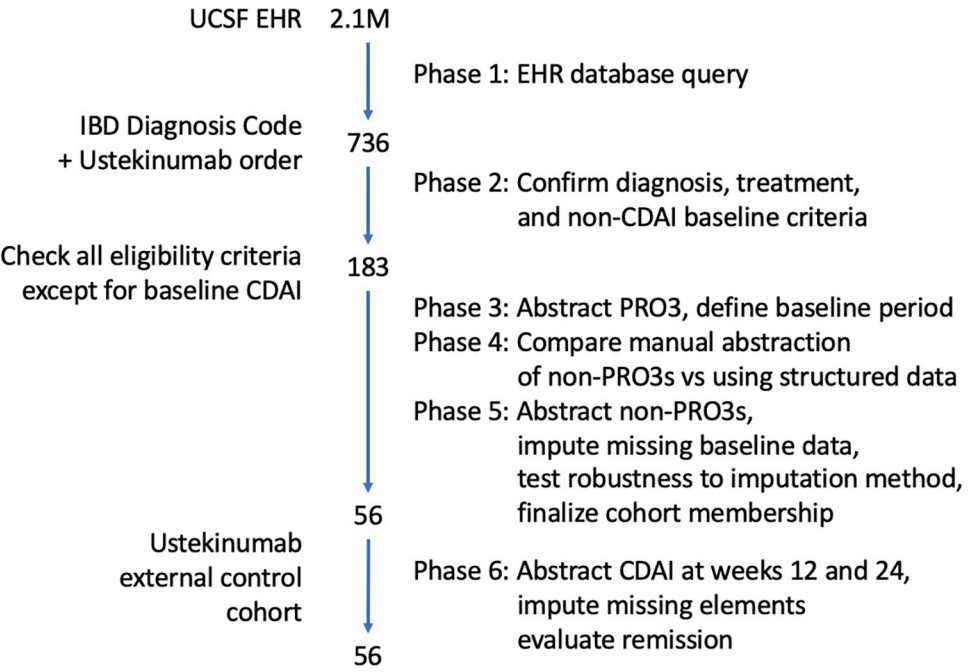

**Fig 1. Study flow diagram.** 736 patient records were screened for inclusion, 56 records met all study criteria. Sequential definition of the cohort described on the left, methods for defining the cohort by study phase summarized on the right.

**Phase 4.**    We abstracted the non-PRO3 elements of the CDAI (e.g. hematocrit, extraintestinal manifestations) using two methods: manual abstraction, and heuristics utilizing structured EHR data like lab values and diagnosis codes (see S3 in S1 File). One of the four chart abstractors completed this on roughly one quarter of the still eligible cohort. We abstracted all non-PRO3 CDAI elements were abstracted at baseline (week -12 to 0) and directly compared the results with the structured data approach.

We performed manual abstraction on a subset of these variables at post-baseline time periods to assess the accuracy of structured data methodology across time. These included the use of antidiarrheals or opiates (binary), the use of steroids (binary), and hematocrit (numeric). Post-baseline time periods corresponded to the times of the primary and secondary endpoints of TRIDENT.

**Phase 5.**    We used the structured data-based method to ascertain all non-PRO3 elements at the baseline period among the 183 patients who met all other criteria (S1 in S1 File). Given missing CDAI elements at baseline, we needed to use imputation to determine which individuals met the baseline CDAI requirement and thus could be confirmed as members of the cohort. We used two random forest-based imputation routines (*MissForest*, *MissRanger*) to assess the sensitivity of cohort membership to the choice of method. Each of these methods utilizes all the other available data train random forest models to impute missing elements, and repeats this process in an iterative fashion with different targets of prediction until the model achieves convergence on all elements [7]. We considered this use of imputation reasonable for two reasons: 1) the included variables corresponded to elements that are directly related to Crohn's disease activity (e.g. CDAI elements, current and prior treatment history, biomarkers), and 2) the dataset contained many of these variables (72 in total), significantly reducing the likelihood of residual bias (see S1 File).

**Phase 6.**    We used *MissForest* to finalize cohort membership using the same method for calculating the CDAI as was used in TRIDENT (S1 File).

## Number of subjects

The sample size calculation in the TRIDENT protocol specified 50 subjects per arm. This study identified and analyzed 56 patients.

## Endpoints

Endpoints included the mean reduction in CDAI at week 12 (TRIDENT primary endpoint) and week 24. Of note, subjects in TRIDENT who entered the study on glucocorticoids were required to remain on them during the induction period. Because real-world clinical practice involves tapering these medications earlier, we included steroid use and steroid-free clinical remission (CDAI $\leq$ 150) at weeks 12 and 24 as additional endpoints.

We used time windows to approximate the true weeks 12 (weeks 10–14) and 24 (weeks 20–28) relative to the date of ustekinumab initiation. We fixed these windows *a priori* based on prior work [5].

We performed manual review to abstract PRO3 elements corresponding to these time windows, and used informatics-based methods to abstract the non-PRO3 elements. We reapplied the *MissForest* algorithm to impute any missing values post-baseline. We analyzed these data to estimate treatment outcomes according to the above endpoints.

## Safety

The assessment of drug safety was not an objective of this pilot study. We did not identify an incidental findings of possible adverse drug events.

## Statistics

This was a descriptive study. Although our original intention was to compare the results of this ECA to that of the target cohort within the TRIDENT trial using statistical hypothesis testing, we did not perform this for a few reasons: 1) we were unable to accurately emulate several major study criteria, specifically pertaining to biologic washouts and stable uses of medications like corticosteroids, and 2) the outcomes of the target cohort from TRIDENT had not been published as of the time that this work was completed. We reported binary outcomes numerically and as a proportion. We reported numeric outcomes by the mean and standard deviation. We performed statistical computing in *R*.

# Results

## Cohort identification

The cohort selection process is outlined in Fig 1 and described according to the phase of the study.

**Phases 1 and 2.** At the time of our EHR database query (April 2020), we identified 736 patients as having an ustekinumab medication order and a Crohn's disease diagnosis code. We manually screened these records to confirm patient eligibility based on the inclusion and exclusionary criteria adapted from TRIDENT (Methods). We found that 526 patients were excluded based on at least one criterion, for example patients who has recently initiated steroids to treat active disease while awaiting their first dose of ustekinumab.

**Phase 3.** We manually abstracted PRO3 elements at timepoints ranging from -16 to 0 weeks relative to starting ustekinumab. We analyzed data availability using different possible time windows at baseline, and identified 12 weeks as the smallest window that did not result in significant missing data at baseline (Fig 2). Data was commonly missing at time points close to the date of ustekinumab induction, reflecting gaps of time between clinic visits where treatments were decided upon and the date that patients actually received intravenous ustekinumab. When clinic visits occurred, we found that all PRO3 elements tended to be documented together. The PROs were more commonly available than lab-based elements such as c-reactive protein (CRP).

**Phase 4.** We abstracted the non-PRO3 elements on a sample of the still eligible cohort (N = 183) using a combination of manual and informatics-based approaches. We found the accuracy of informatic approaches to abstracting the CDAI to be high compared to a gold-standard of manual review (Fig 3). The degree of agreement, measured by Pearson's $r^2$, ranged from 0.91 to 0.96 across timepoints (Tables 1–3). This high correlation appeared to be driven by the fact that for most patients, the major contributor to the total CDAI came from PRO3 elements rather than the other elements ascertained by informatics-based methods.

**Phase 5.** 30% of the cohort was missing at least one PRO3 element at baseline (Table 4). To handle this missing data and determine cohort membership, we compared two methods (*MissRanger*, *MissForest*) for performing single imputation. These random forest-based methods differ in their modes of optimization as well as their final models, which are fit according to a stochastic process. The two models gave very similar results relevant to the selection of the baseline cohort and their outcomes (Table 5).

**Phase 6.** We selected *MissForest* as the method for imputing the baseline data and thereby finalize the cohort of 56 patients meeting all study criteria (Table 6). We abstracted post-baseline PRO3 elements in these patients, and used the previously described informatics algorithm to abstract all other post-baseline variables. To handle post-baseline missing data (Table 7), we reapplied the *MissForest* algorithm to complete the dataset across timepoints, enabling us to assess the outcomes of this cohort.

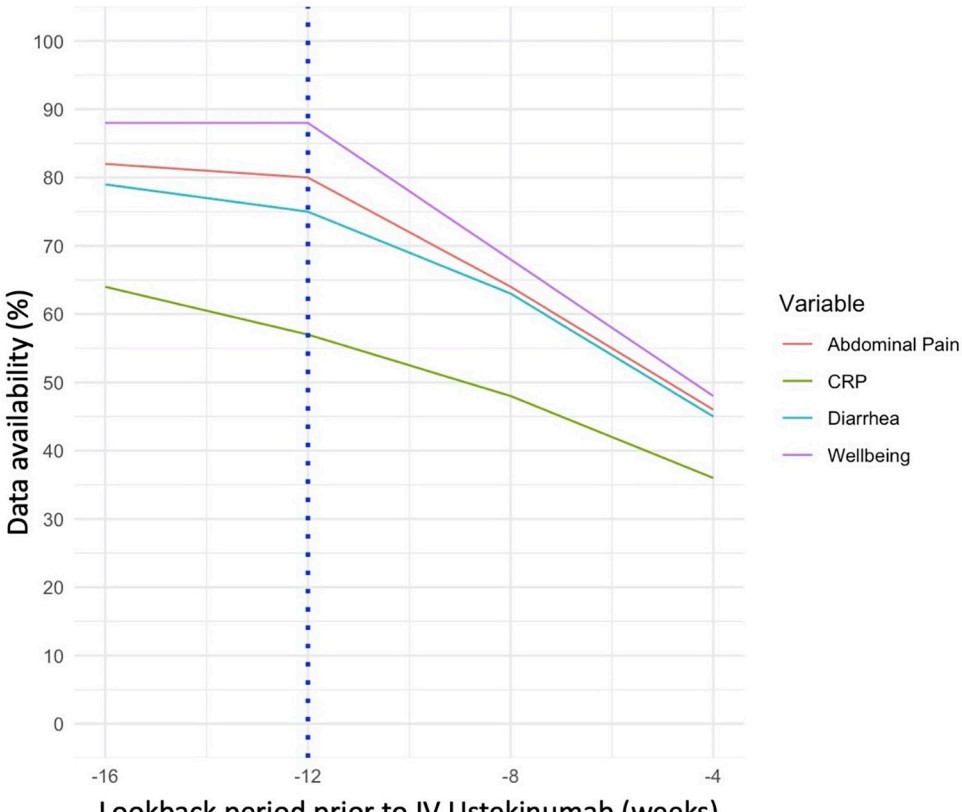

**Fig 2. Impact of different definitions for the baseline time period on variable availability.** These results correspond to the 183 patients who were not excluded by non-CDAI related study criteria. All of these patient records underwent manual review to ascertain the presence of PROs occurring during the 16 week period prior to the first dose of ustekinumab (week 0). The blue dotted line corresponds to the cut point (12 week lookback) that was selected following a visual review of the trends depicted here. CRP corresponds to c-reactive protein.

## Endpoints

Ustekinumab was associated with a 95 point mean reduction in the CDAI by week 12, and a 133 point reduction by week 24 (N = 56; Fig 4, Table 8). The proportions of patients in steroid-free clinical remission were 23% and 34% at Weeks 12 and 24 respectively. 38% of the cohort was using steroids at the baseline timepoint. Out of the cohort of 56, 7 (13%) and 9 (16%) remained on steroids at weeks 12 and 24 respectively (Table 9).

## Discussion

We used a combination of manual review, informatics, and imputation to pilot a method for creating external control arms in Crohn's disease. We applied this method to identify a real-world cohort resembling the ustekinumab arm in TRIDENT, a recently completed phase 2b trial. We found that algorithms utilizing structured EHR data were accurate at ascertaining the CDAI (both components and in aggregate) and may be a favorable alternative to manual review for non-PRO3 components. We found a substantial amount of missing data in the context of retrospective use for this study design. However, our results suggested that different imputation models may be equivalent in their impacts on cohort definition and outcome measurement. Lastly, this observational cohort appeared to demonstrate a plausible improvement

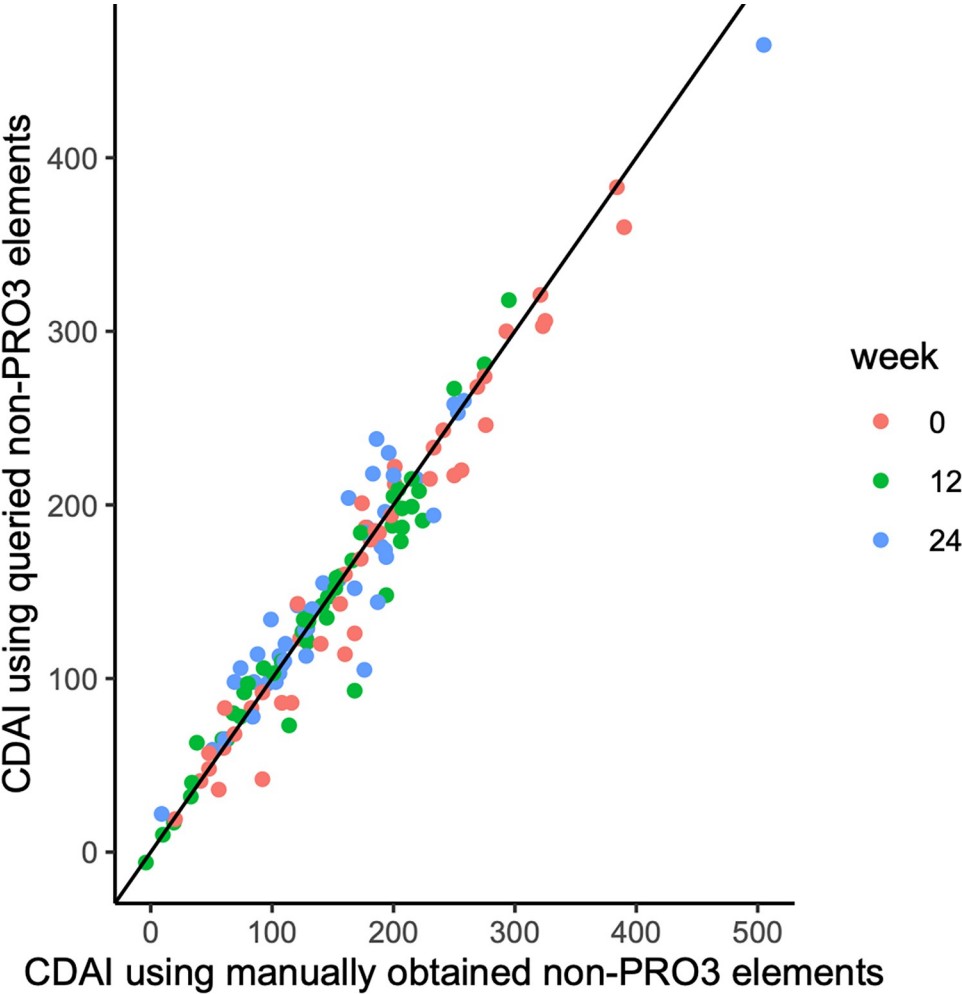

**Fig 3. Comparison of total CDAI as computed using CDAI elements ascertained informatically vs manually.** The
y-axis corresponds to the CDAI calculated by database query of non-PRO3 elements. The x-axis corresponds to the
CDAI calculated by manual abstraction.

in disease activity by several measures, consistent with the well-established efficacy of usteki-
numab [8]. Our results suggest that roughly a third of the ustekinumab-treated cohort of TRI-
DENT will be in steroid-free remission at week 24. The actual outcome of that cohort is
pending the publication of the TRIDENT study.

Interest in external control arm studies has continued to grow in recent years. This has
directly followed from several trends: 1) the increasing availability of large clinical datasets
such as from medical claims and EHRs, 2) advances in methods for organizing and extracting

**Table 1. Comparison of total CDAI as computed using CDAI elements ascertained informatically vs manually.**  All results correspond to the results of manual
abstraction by one chart reviewer (45 patients).

| Time | Mean Absolute Error | Pearson's $r^2$ | Steroid-free Remission (manual) | Steroid-free Remission (informatics) |
|---|---|---|---|---|
| Baseline | 13 | 0.96 | 0/14 | 0/14 |
| Week 12 | 12 | 0.93 | 5/14 | 6/13 |
| Week 24 | 17 | 0.91 | 6/14 | 5/13 |

**Table 2. Comparison of the results of non-PRO3, binary CDAI elements by manual vs informatics methods.**
Accuracy is reported relative to manually abstracted data (gold-standard). W12 and W24 correspond to the week 12 and 24 periods respectively. Comparisons were made against the annotations performed by one chart reviewer (45 patients), roughly a quarter of the 183 patients who met all eligibility criteria prior to application of the baseline CDAI requirement.

| Variable | Time | Accuracy |
|---|---|---|
| Fever | Baseline | 1 |
| Arthritis | Baseline | 0.93 |
| Uveitis | Baseline | 0.98 |
| Anal Fissure/Fistula/Abscess | Baseline | 0.98 |
| Erythema Nodosum, Pyoderma Gangrenosum, Aphthous Stomatitis | Baseline | 0.96 |
| Other Fistula | Baseline | 1 |
| Antidiarrheal or opiate use | Baseline | 0.91 |
| Antidiarrheal or opiate use | W12 | 0.96 |
| Antidiarrheal or opiate use | W24 | 0.91 |

information from these data, 3) the large and rising costs of randomized trials [9], and 4) increasingly favorable attitudes by regulators towards their use [2].

If done well, these studies have the potential to transform the way we generate clinical evidence. They can inform the safety and efficacy of existing therapies, as well as new ones by indirect comparison. They may also help answer questions about comparative effectiveness, cost-benefits, and precision medicine, particularly in cohorts who might not have been studied in registrational trials.

Despite this potential, our study underscores important differences between this variety of retrospective research and their prospective counterparts. It is more difficult to create a high-quality external control arm for protocols that 1) specify exact study visit timing, 2) prioritize specific outcome measurements that are not commonly obtained in ordinary practice, and 3) constrain what treatments a patient may or may not receive (deviating from clinical care).

The principal limitation of our methodology was missing data. This problem can be understood as the natural consequence of retrospective deviation from prospective study design in three ways.

## Study timing

The timing of real-world clinic visits follows clinical necessity as well as the individual preferences of providers and patients. It is not uncommon for patients to have one clinic visit that determines the need to start a new therapy, and another to evaluate treatment response. Delays between the decision to start treatment and the receipt of treatment in the real-world essentially guarantees missing data at the study visit equivalent of week 0. These delays are particularly magnified in IBD as treated in the US, where sick patients commonly need expensive biologics that require payor authorization and scheduling infusions.

**Table 3. Comparison of the results of non-PRO3, continuous CDAI elements by manual vs informatics methods.** Mean absolute error was calculated only for values that were non-missing by both informatics and manual methods. Comparisons were made against the annotations performed by one chart reviewer (45 patients), roughly a quarter of the 183 patients who met all eligibility criteria prior to application of the baseline CDAI requirement.

| Variable | Time | Mean Absolute Error | Missing data (informatics) | Missing data (manual) |
|---|---|---|---|---|
| Weight | Baseline | 0.1 | 9% | 2% |
| Hematocrit | Baseline | 0.1 | 40% | 20% |
| Hematocrit | W12 | 0 | 78% | 73% |
| Hematocrit | W24 | 0.2 | 78% | 67% |

**Table 4. Characterization of missing PRO3 elements at baseline.** Proportions correspond to the 183 patients who met all eligibility criteria prior to application of the baseline CDAI requirement.

| Variable | Missing PRO3 data |
|---|---|
| Abdominal Pain | 27% |
| Diarrhea | 30% |
| Wellbeing | 20% |

This situation is similar at the time of follow-up. The precise timing of a follow-up visit can differ based on several factors, including provider availability, patient preferences, and even what drug a patient receives (which in turn informs the most reasonable time to expect a response).

Our study attempted to overcome these differences. We noted the high presence of missing clinic visits at week 0, and therefore used an empirical calibration approach to identify the time window for estimating a patient's actual CDAI at the time of ustekinumab induction. We use predefined windows of ± 2 and 4 weeks for the outcome assessment for similar reasons, to avoid the retrospective miscalibration of clinic visit timings with that of TRIDENT. Future studies are needed to explore the use of other patient-interactive technologies, such as timed patient surveys embedded into EHR systems, to better address this limitation.

## Outcome measures

We found that the informal capture of clinical data can deviate substantially from that of common clinical trial instruments. The CDAI requires a significant amount of data collection across a wide variety of domains–PROs, vitals, laboratories, and extraintestinal diagnoses. It also requires week-long symptom diaries. Unsurprisingly, the CDAI has had poor uptake in actual clinical practice. This was a large driver of missing data in this study.

Our findings suggest substantial potential for simplifying these indices, or more generally, approaches to better align them to the realities of real-world practice. A large part of the reason why our algorithms were as accurate as they were for ascertaining the CDAI was because of class imbalance. That is, most patients did not have any EHR-based evidence for extraintestinal manifestations, and thus their CDAIs were strongly driven by the PROs (all of which were abstracted manually). A further simplification of the PROs from multi-level ordinal variables to even a binary variable might make for a good tradeoff between precision and suitability for routine clinical capture. Future work is needed to develop 'real-world ready' instruments that maintain responsiveness and validity.

## Constrained treatments

The final point, getting at the heart of the difference between clinical care and controlled studies, resulted in a different kind of missing data problem: one of diminished sample size rather

**Table 5. Comparison of the results of two imputation models.** Imputation models were applied to all of the otherwise eligible patients (less the baseline CDAI requirement) that had been assigned to one chart abstractor (45 patients).

| Measure | Count/ Proportion (MissForest) | Count/ Proportion (MissRanger) |
|---|---|---|
| Patients with baseline CDAIs within range (220–450) | 12/45 (27%) | 13/45 (29%) |
| Patients in remission (CDAI ≤ 150) at week 12 among those with within range baseline CDAIs | 5/12 (42%) | 5/13 (38%) |
| Patients in remission (CDAI ≤ 150) at week 24 among those with within range baseline CDAIs | 5/12 (42%) | 5/13 (38%) |

**Table 6. Characterization of the study cohort.** *: Reported as median, other continuous variables reported as mean ± standard deviation. Parenthetical modifiers to the disease location elements correspond to the Montreal Classification of Crohn's disease.

| | UCSF External Control Arm (N = 56) |
|---|---|
| Age in years | 34 ± 14 |
| Female Gender | 59% |
| Disease Duration | 10 ± 11 |
| C-Reactive Protein* | 12.3 |
| History of Biologic Intolerance or Refractoriness | 89% |
| Disease Location: | |
| • Ileal (L1) | 13% |
| • Ileocolonic (L3) | 52% |
| • Colonic (L2) | 30% |
| • Other | 5% |
| Baseline CDAI | 305 ± 51 |
| Baseline steroid use | 38% |

than missing values. Although we began this study with 736 potential candidates, we excluded 75% of this population after sequentially applying the major eligibility criteria used to screen subjects in TRIDENT. This study was not designed to measure what proportion of candidates were excluded by different criteria. We applied a 'greedy' selection approach (eliminating candidates at the earliest evidence of disqualification) to avoid the labor of abstracting 7,360 data points (10 non-CDAI eligibility criteria).

However, our impression was that many patients were disqualified due to changes in therapy during the washout period prior to the date of ustekinumab induction. This of course is quite natural: patients with active Crohn's disease who are under clinical care are highly likely to undergo changes in treatment, whether rapid transitions from prior treatment to new ones, or the addition of adjunctive/bridging agents like steroids. This was our reason for removing the biologic washout requirement as an eligibility criterion in this study.

This misalignment between experiments designed to measure treatment effects and clinical practice designed to treat patients results in a significant loss in study efficiency. One solution might involve using larger clinical datasets, to find many more of those rare symptomatic patients who ordinarily would have been treated but by chance were not. However, this approach may increase the risks of unmeasured confounding and residual bias. A potentially better solution would be that of an EHR-enabled registry. The use of phenotyping algorithms

**Table 7. Characterization of missing PRO3 elements across all time periods.** Proportions correspond to all 56 members of the external cohort.

| Variable | Time | Missing PRO3 data |
|---|---|---|
| Abdominal Pain | Baseline | 25% |
| Diarrhea | Baseline | 27% |
| Wellbeing | Baseline | 13% |
| Abdominal Pain | Week 12 | 59% |
| Diarrhea | Week 12 | 77% |
| Wellbeing | Week 12 | 46% |
| Abdominal Pain | Week 24 | 61% |
| Diarrhea | Week 24 | 66% |
| Wellbeing | Week 24 | 43% |

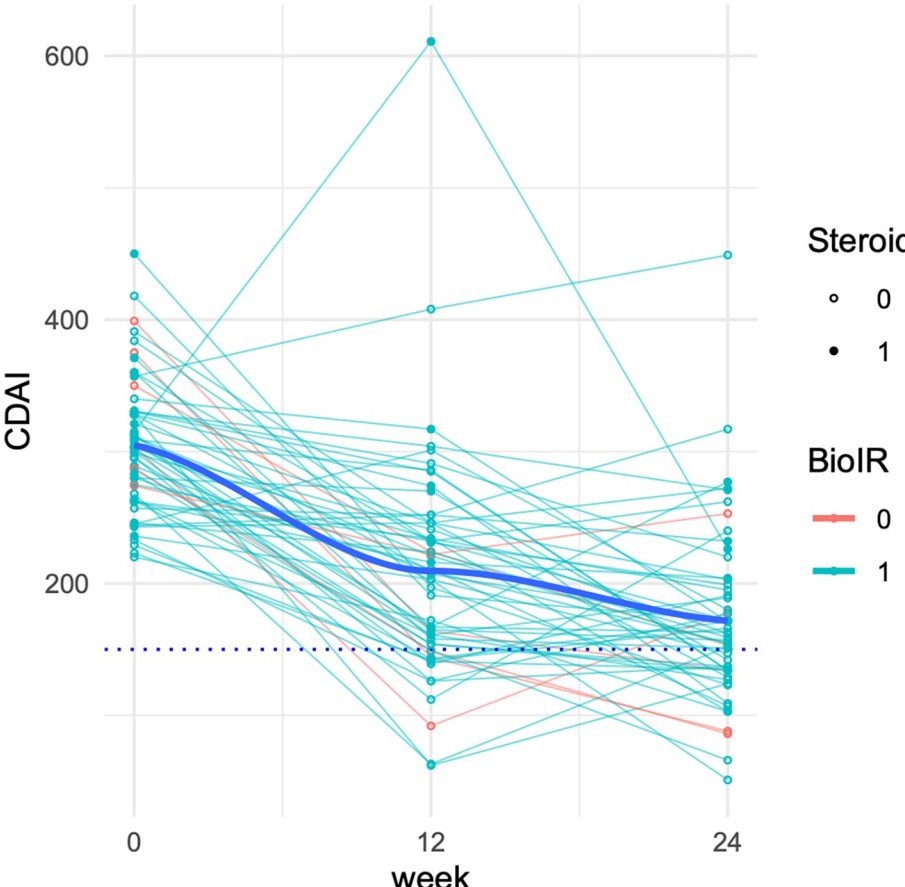

**Fig 4. Change in the Crohn's Disease Activity Index (CDAI) over time.** Lines correspond to individual patient trajectories, where turquoise corresponds to patients who were biologic-intolerant or refractory (BioIR) and red corresponds to patients who were biologic-naïve. Open circles represent time points where the patient was not receiving oral steroids whereas filled circles represent the use oral steroids at a given time.

to screen patients prior to manual review might be a more cost-effective way to efficiently recruit patients, guarantee the timing and capture of relevant data, and involve more under-represented patients in studies that culminate in practice-changing evidence.

Strengths of this study include the use of predefined chart review protocol, validation of algorithms against that of manual review, sensitivity analyses to explore the effects of various design decisions on outcomes, the release of our raw data and code, and the identification of treatment effects that are broadly consistent the literature. Weaknesses as described above pertain primarily to missing data and the resulting inability to make stronger inferences about treatment effectiveness. We additionally note that this was a single-center study: the

**Table 8. Efficacy endpoints.**

| Outcome | Numerical Result [Mean ± SE, N/D (%)] |
| --- | --- |
| CDAI Reduction by Week 12 | 95 ± 13 |
| CDAI Reduction by Week 24 | 133 ± 11 |
| Proportion in steroid-free remission (CDAI ≤ 150) at Week 12 | 13/56 (23%) |
| Proportion in steroid-free remission (CDAI ≤ 150) at Week 24 | 19/56 (34%) |

**Table 9. Steroid use across timepoints.**

| Time | Proportion (%) |
|------|----------------|
| Baseline | 21/56 (37%) |
| Week 12 | 7/56 (13%) |
| Week 24 | 9/56 (16%) |

generalizability of this methodology to other centers with potentially different patient populations and data quality remain to be seen in future work.

In conclusion, we have piloted an approach for performing an external control arm study in Crohn's disease. Future studies are needed to improve the alignment between prospective study design and real-world clinical care in complex diseases such as Crohn's disease.

## Supporting information

**S1 Checklist.**
(PDF)

**S1 File. Document containing 1) Definitions of informatically-ascertained CDAI elements, 2) Calculation of the CDAI, 3) Abstraction definitions.**
(DOCX)

**S1 Data. Zipped files containing data and analytical code.**
(ZIP)

## Acknowledgments

The authors thank Jennifer Creasman, UCSF Academic Research Services, and Clinical Data Research Consultation services for clinical informatics support.

## Author Contributions

**Conceptualization:** Vivek A. Rudrapatna.

**Data curation:** Vivek A. Rudrapatna, Yao-Wen Cheng, Colin Feuille, Arman Mosenia, Jonathan Shih, Yongmei Shi.

**Formal analysis:** Vivek A. Rudrapatna, Yongmei Shi.

**Funding acquisition:** Vivek A. Rudrapatna, Atul J. Butte, Najat S. Khan, Benjamin D. Martini.

**Investigation:** Vivek A. Rudrapatna.

**Methodology:** Vivek A. Rudrapatna, Yongmei Shi.

**Project administration:** Vivek A. Rudrapatna, Olivia Roberson, Benjamin Rubin.

**Resources:** Vivek A. Rudrapatna.

**Software:** Vivek A. Rudrapatna, Yongmei Shi.

**Supervision:** Vivek A. Rudrapatna, Uma Mahadevan, Nicholas Skomrock, Ngozi Erondu, Christel Chehoud, Saquib Rahim, David Apfel, Mark Curran, Christopher O'Brien, Natalie Terry, Benjamin D. Martini.

**Validation:** Vivek A. Rudrapatna.

**Visualization:** Vivek A. Rudrapatna.

**Writing – original draft:** Vivek A. Rudrapatna, Yongmei Shi.

**Writing – review & editing:** Vivek A. Rudrapatna, Yao-Wen Cheng, Colin Feuille, Arman Mosenia, Atul J. Butte, Uma Mahadevan, Nicholas Skomrock, Ngozi Erondu, Christel Chehoud, Saquib Rahim, David Apfel, Christopher O'Brien, Natalie Terry, Benjamin D. Martini.

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
