## [Decision Letter · Decision Letter 0]

25 Nov 2022

PONE-D-22-27576Creation of an ustekinumab external control arm for Crohn’s disease using electronic health records data: a pilot studyPLOS ONE

Dear Dr. Rudrapatna,

Thank you for submitting your manuscript to PLOS ONE. After careful consideration, we feel that it has merit but does not fully meet PLOS ONE’s publication criteria as it currently stands. Therefore, we invite you to submit a revised version of the manuscript that addresses the points raised during the review process. Please submit your revised manuscript by Jan 09 2023 11:59PM. If you will need more time than this to complete your revisions, please reply to this message or contact the journal office at plosone@plos.org. Please include the following items when submitting your revised manuscript:A rebuttal letter that responds to each point raised by the academic editor and reviewer(s). You should upload this letter as a separate file labeled 'Response to Reviewers'.A marked-up copy of your manuscript that highlights changes made to the original version. You should upload this as a separate file labeled 'Revised Manuscript with Track Changes'.An unmarked version of your revised paper without tracked changes. You should upload this as a separate file labeled 'Manuscript'.

We look forward to receiving your revised manuscript.

Kind regards,

Sreeram V. Ramagopalan

Academic Editor

PLOS ONE

Journal Requirements:

"VR received funding from Janssen Research and Development LLC (www.janssen.com).

VR received funding from the UCSF Division of Gastroenterology (gastroenterology.ucsf.edu).

VR and AJB received funding from UCSF Bakar Computational Health Sciences Institute (bakarinstitute.ucsf.edu)

The contents of this manuscript are solely the responsibility of the authors and do not necessarily represent the official views of the NIH. None of the study sponsors had any influence over the data interpretation or conclusions from this study."

Please state what role the funders took in the study.  If the funders had no role, please state: ""The funders had no role in study design, data collection and analysis, decision to publish, or preparation of the manuscript."" If this statement is not correct you must amend it as needed. 

"NS, NE, CC, SR, DA, MC, NSK, CO, NT, and BDM are employees of Janssen Pharmaceuticals, a for-profit entity that owns all rights to the drug ustekinumab. UM is a consultant for Janssen Pharmaceuticals. AJB is a co-founder and consultant to Personalis and NuMedii; consultant to Mango Tree Corporation, and in the recent past, Samsung, 10x Genomics, Helix, Pathway Genomics, and Verinata (Illumina); has served on paid advisory panels or boards for Geisinger Health, Regenstrief Institute, Gerson Lehman Group, AlphaSights, Covance, Novartis, Genentech, and Merck, and Roche; is a shareholder in Personalis and NuMedii; is a minor shareholder in Apple, Meta (Facebook), Alphabet (Google), Microsoft, Amazon, Snap, 10x Genomics, Illumina, Regeneron, Sanofi, Pfizer, Royalty Pharma, Moderna, Sutro, Doximity, BioNtech, Invitae, Pacific Biosciences, Editas Medicine, Nuna Health, Assay Depot, and Vet24seven, and several other non-health related companies and mutual funds; and has received honoraria and travel reimbursement for invited talks from Johnson and Johnson, Roche, Genentech, Pfizer, Merck, Lilly, Takeda, Varian, Mars, Siemens, Optum, Abbott, Celgene, AstraZeneca, AbbVie, Westat, and many academic institutions, medical or disease specific foundations and associations, and health systems.  Atul Butte receives royalty payments through Stanford University, for several patents and other disclosures licensed to NuMedii and Personalis.  Atul Butte’s research has been funded by NIH, Peraton (as the prime on an NIH contract), Genentech, Johnson and Johnson, FDA, Robert Wood Johnson Foundation, Leon Lowenstein Foundation, Intervalien Foundation, Priscilla Chan and Mark Zuckerberg, the Barbara and Gerson Bakar Foundation, and in the recent past, the March of Dimes, Juvenile Diabetes Research Foundation, California Governor’s Office of Planning and Research, California Institute for Regenerative Medicine, L’Oreal, and Progenity. The authors have declared that no competing interests exist."

We note that one or more of the authors are employed by a commercial company: Janssen Pharmaceuticals, Personalis and NuMedii, etc.

(2) Please also provide an updated Competing Interests Statement declaring this commercial affiliation along with any other relevant declarations relating to employment, consultancy, patents, products in development, or marketed products, etc.  

Within your Competing Interests Statement, please confirm that this commercial affiliation does not alter your adherence to all PLOS ONE policies on sharing data and materials by including the following statement: ""This does not alter our adherence to  PLOS ONE policies on sharing data and materials.” (as detailed online in our guide for authors http://journals.plos.org/plosone/s/competing-interests) . 

If this adherence statement is not accurate and  there are restrictions on sharing of data and/or materials, please state these. Please note that we cannot proceed with consideration of your article until this information has been declared.

6. Please amend the manuscript submission data (via Edit Submission) to include authors: Colin Feuille MD, Yongmei Shi PhD

7. Please amend either the abstract on the online submission form (via Edit Submission) or the abstract in the manuscript so that they are identical.

8. Please upload a copy of Figure 8 to which you refer in your text on page 26. If the figure is no longer to be included as part of the submission please remove all reference to it within the text.

9. We note you have included a table to which you do not refer in the text of your manuscript. Please ensure that you refer to Table 8 in your text; if accepted, production will need this reference to link the reader to the Table.

10. Please include captions for your Supporting Information files at the end of your manuscript, and update any in-text citations to match accordingly. Please see our Supporting Information guidelines for more information: http://journals.plos.org/plosone/s/supporting-information. 

Reviewers' comments:

Reviewer's Responses to Questions

**Comments to the Author**

1. Is the manuscript technically sound, and do the data support the conclusions?

Reviewer #1: Partly

2. Has the statistical analysis been performed appropriately and rigorously? 

Reviewer #1: I Don't Know

3. Have the authors made all data underlying the findings in their manuscript fully available?

Reviewer #1: Yes

4. Is the manuscript presented in an intelligible fashion and written in standard English?

Reviewer #1: Yes

5. Review Comments to the Author

Reviewer #1: The authors made an effort to emulate the control arm of a clinical trial in Crohn's disease. I have two major concerns with this manuscript.

First, the concept behind is either contentious or not clear. Starting with the growing accepted concept that external control arms are needed in many situations like for example rare disease and many oncology settings, the paper's motivation appears to be that this concept should be extended to more common diseases because of the cost and complexity of running trials. I think this is very contentious. Randomization still has a value and should be pursued as much as possible. It does not seem justified that external control arms where clinical trials are feasible are justified. But the authors also appear to deviate from this concept in some other parts of the paper, as the concept of running this feasibility exercise is more loosely justified in terms of replacing "prospective" studies, and in some other places their rationale is about "complementing" other types of research. There should be clarity on the place where the authors see this. If the main justification is about replacing a phase III RCT for this kind of approach, I do not think it is a good scientific approach. If the idea is about post laungh evidence generation, or about adding to the totality of the evidence regarding new comparators, I think it will be more justitiable.

Analytically. the authors made a good effort but it falls short on many elements.

First, they did not fully replicate the arm in question, two exclusion criteria were left out. I do not know how the control arms can be emulated. It is not clear if this was an a priori or ex-post decision. Did they try first with all the criteria and numbers were too low or was this an a priori plan? And if it is an a priory plan, how can it emulate the control arm of the trial if these criteria were left out. If the interest was about emulating the control arm, we do not know if a full emulation is possible. Second, to show empirically that the emulation was close, they should present results comparing versus the trial. Actually they should have matched patients using trial data. But all these comparisons or analytical approaches are not in scope and are important. Finally, this is a single center attempt, we do not know if this approach would work for other centers. How dependent on the data quality of the center was this approach? How would it vary if attempting to do a multi-center emulation.

In conclusion, my advice is to deeply clarify the conceptual intent about the role of this exercise and provide more analyses that use the trial baseline information. In addition, generalizability should be explored. The paper, as is, minimally needs to clarify its conceptual intent and recognize analytical limitations. And in that case, it may be a better fit for a medical informatics journal.

6. PLOS authors have the option to publish the peer review history of their article (what does this mean?). If published, this will include your full peer review and any attached files.

Reviewer #1: No

---

## [Author Response · Author response to Decision Letter 0]

28 Dec 2022

December 15, 2022

To the editor and reviewer:

Thank you for your review of our manuscript. We have a number of revisions in response to your feedback. Details and response to your comments are below in green font and preceeded by double asterisks.

ACADEMIC EDITOR COMMENTS:

**Thank you, this has now been done.

**Thank you, now added.

**Now done

"VR received funding from Janssen Research and Development LLC (www.janssen.com).

VR received funding from the UCSF Division of Gastroenterology (gastroenterology.ucsf.edu).

VR and AJB received funding from UCSF Bakar Computational Health Sciences Institute (bakarinstitute.ucsf.edu)

The contents of this manuscript are solely the responsibility of the authors and do not necessarily represent the official views of the NIH. None of the study sponsors had any influence over the data interpretation or conclusions from this study."

Please state what role the funders took in the study. If the funders had no role, please state: ""The funders had no role in study design, data collection and analysis, decision to publish, or preparation of the manuscript."" If this statement is not correct you must amend it as needed. 

**Now included in the revised cover letter

"NS, NE, CC, SR, DA, MC, NSK, CO, NT, and BDM are employees of Janssen Pharmaceuticals, a for-profit entity that owns all rights to the drug ustekinumab. UM is a consultant for Janssen Pharmaceuticals. AJB is a co-founder and consultant to Personalis and NuMedii; consultant to Mango Tree Corporation, and in the recent past, Samsung, 10x Genomics, Helix, Pathway Genomics, and Verinata (Illumina); has served on paid advisory panels or boards for Geisinger Health, Regenstrief Institute, Gerson Lehman Group, AlphaSights, Covance, Novartis, Genentech, and Merck, and Roche; is a shareholder in Personalis and NuMedii; is a minor shareholder in Apple, Meta (Facebook), Alphabet (Google), Microsoft, Amazon, Snap, 10x Genomics, Illumina, Regeneron, Sanofi, Pfizer, Royalty Pharma, Moderna, Sutro, Doximity, BioNtech, Invitae, Pacific Biosciences, Editas Medicine, Nuna Health, Assay Depot, and Vet24seven, and several other non-health related companies and mutual funds; and has received honoraria and travel reimbursement for invited talks from Johnson and Johnson, Roche, Genentech, Pfizer, Merck, Lilly, Takeda, Varian, Mars, Siemens, Optum, Abbott, Celgene, AstraZeneca, AbbVie, Westat, and many academic institutions, medical or disease specific foundations and associations, and health systems. Atul Butte receives royalty payments through Stanford University, for several patents and other disclosures licensed to NuMedii and Personalis. Atul Butte’s research has been funded by NIH, Peraton (as the prime on an NIH contract), Genentech, Johnson and Johnson, FDA, Robert Wood Johnson Foundation, Leon Lowenstein Foundation, Intervalien Foundation, Priscilla Chan and Mark Zuckerberg, the Barbara and Gerson Bakar Foundation, and in the recent past, the March of Dimes, Juvenile Diabetes Research Foundation, California Governor’s Office of Planning and Research, California Institute for Regenerative Medicine, L’Oreal, and Progenity. The authors have declared that no competing interests exist."

We note that one or more of the authors are employed by a commercial company: Janssen Pharmaceuticals, Personalis and NuMedii, etc.

**I have added this text to the section “Funding Statement”.

(2) Please also provide an updated Competing Interests Statement declaring this commercial affiliation along with any other relevant declarations relating to employment, consultancy, patents, products in development, or marketed products, etc. 

Within your Competing Interests Statement, please confirm that this commercial affiliation does not alter your adherence to all PLOS ONE policies on sharing data and materials by including the following statement: ""This does not alter our adherence to PLOS ONE policies on sharing data and materials.” (as detailed online in our guide for authors http://journals.plos.org/plosone/s/competing-interests) . 

If this adherence statement is not accurate and there are restrictions on sharing of data and/or materials, please state these. Please note that we cannot proceed with consideration of your article until this information has been declared.

**Now revised and additionally included in the cover letter.

6. Please amend the manuscript submission data (via Edit Submission) to include authors: Colin Feuille MD, Yongmei Shi PhD

**Now done.

7. Please amend either the abstract on the online submission form (via Edit Submission) or the abstract in the manuscript so that they are identical.

**Now done.

8. Please upload a copy of Figure 8 to which you refer in your text on page 26. If the figure is no longer to be included as part of the submission please remove all reference to it within the text.

**Sorry – this was an error, meant to say Figure 4. Now corrected.

9. We note you have included a table to which you do not refer in the text of your manuscript. Please ensure that you refer to Table 8 in your text; if accepted, production will need this reference to link the reader to the Table.

**Sorry – this was an error, meant to say Table 8. Now corrected.

10. Please include captions for your Supporting Information files at the end of your manuscript, and update any in-text citations to match accordingly. Please see our Supporting Information guidelines for more information: http://journals.plos.org/plosone/s/supporting-information. 

**Done

REVIEWER’S COMMENTS

Reviewer #1: The authors made an effort to emulate the control arm of a clinical trial in Crohn's disease. I have two major concerns with this manuscript.

First, the concept behind is either contentious or not clear. Starting with the growing accepted concept that external control arms are needed in many situations like for example rare disease and many oncology settings, the paper's motivation appears to be that this concept should be extended to more common diseases because of the cost and complexity of running trials. I think this is very contentious. Randomization still has a value and should be pursued as much as possible. It does not seem justified that external control arms where clinical trials are feasible are justified. But the authors also appear to deviate from this concept in some other parts of the paper, as the concept of running this feasibility exercise is more loosely justified in terms of replacing "prospective" studies, and in some other places their rationale is about "complementing" other types of research. There should be clarity on the place where the authors see this. If the main justification is about replacing a phase III RCT for this kind of approach, I do not think it is a good scientific approach. If the idea is about post laungh evidence generation, or about adding to the totality of the evidence regarding new comparators, I think it will be more justitiable.

**Thank you for your feedback. We did not intend to imply that ECAs should replace phase 3 RCTs anytime in the near future, particularly for diseases like Crohn’s disease where these high-quality studies continue to be relatively feasible and where current retrospective emulation methods (as we show here) are clearly very limited. Although we can see how the last version of our manuscript could give this impression.

**That said, we do think that this field needs to continue to develop methods for using EHR data in clinical research and to benchmark retrospective studies against prospective ones. Even though our pilot study suffered from many limitations, we think that many readers can learn from our experience, anticipate and avoid certain missteps, improve the quality of retrospective evidence, and address the occasional evidence gaps that lie between controlled environments and routine clinical care. 

**We have modified the language throughout to reflect this, including the abstract (lines 24-27), introduction (revised paragraph 1), methods (line 89, 216-220), and the discussion.

Analytically. the authors made a good effort but it falls short on many elements.

First, they did not fully replicate the arm in question, two exclusion criteria were left out. I do not know how the control arms can be emulated. It is not clear if this was an a priori or ex-post decision. Did they try first with all the criteria and numbers were too low or was this an a priori plan? And if it is an a priory plan, how can it emulate the control arm of the trial if these criteria were left out. If the interest was about emulating the control arm, we do not know if a full emulation is possible. Second, to show empirically that the emulation was close, they should present results comparing versus the trial. Actually they should have matched patients using trial data. But all these comparisons or analytical approaches are not in scope and are important. Finally, this is a single center attempt, we do not know if this approach would work for other centers. How dependent on the data quality of the center was this approach? How would it vary if attempting to do a multi-center emulation.

**Thank you for your feedback. We have made revisions to more clearly acknowledge these limitations:

-Lines 125-128 now make clear that the decisions to leave the two exclusion criteria out were made ex post, and what the reasons were. 

-Lines 216-220 reinforce the above and indicate the reasons for not statistically comparing the outcomes of our cohort that of TRIDENT.

-Lines 435-437 emphasize the limitations of this single-center effort and the possibility that our method may not generalize to other centers.

In conclusion, my advice is to deeply clarify the conceptual intent about the role of this exercise and provide more analyses that use the trial baseline information. In addition, generalizability should be explored. The paper, as is, minimally needs to clarify its conceptual intent and recognize analytical limitations. And in that case, it may be a better fit for a medical informatics journal.

**As above, we do agree with you. We have made several revisions of the text and hope that it addresses the important points you raised above. Although we considered a medical informatics journal as a potential venue, we think that PLOS ONE has a wider readership and would allow us to communicate these findings with more members of our target audience.

---

## [Decision Letter · Decision Letter 1]

13 Feb 2023

Creation of an ustekinumab external control arm for Crohn’s disease using electronic health records data: a pilot study

PONE-D-22-27576R1

Dear Dr. Rudrapatna,

We’re pleased to inform you that your manuscript has been judged scientifically suitable for publication and will be formally accepted for publication once it meets all outstanding technical requirements.

Kind regards,

Sreeram V. Ramagopalan

Academic Editor

PLOS ONE

Additional Editor Comments (optional):

Reviewers' comments:

Reviewer's Responses to Questions

**Comments to the Author**

1. If the authors have adequately addressed your comments raised in a previous round of review and you feel that this manuscript is now acceptable for publication, you may indicate that here to bypass the “Comments to the Author” section, enter your conflict of interest statement in the “Confidential to Editor” section, and submit your "Accept" recommendation.

Reviewer #1: All comments have been addressed

2. Is the manuscript technically sound, and do the data support the conclusions?

Reviewer #1: Yes

3. Has the statistical analysis been performed appropriately and rigorously? 

Reviewer #1: Yes

4. Have the authors made all data underlying the findings in their manuscript fully available?

Reviewer #1: Yes

5. Is the manuscript presented in an intelligible fashion and written in standard English?

Reviewer #1: Yes

6. Review Comments to the Author

Reviewer #1: The authors addressed the comments transparently and the points in question are communicated clearly to the readers. The manuscript is in my opinion suitable for publication.

7. PLOS authors have the option to publish the peer review history of their article (what does this mean?). If published, this will include your full peer review and any attached files.

Reviewer #1: **Yes: **Gerardo Machnicki

---

## [Editor Report · Acceptance letter]

20 Feb 2023

PONE-D-22-27576R1 

Creation of an ustekinumab external control arm for Crohn’s disease using electronic health records data: a pilot study 

Dear Dr. Rudrapatna:

I'm pleased to inform you that your manuscript has been deemed suitable for publication in PLOS ONE. Congratulations! Your manuscript is now with our production department. 

Kind regards, 

on behalf of

Dr. Sreeram V. Ramagopalan 

Academic Editor

PLOS ONE